# How Do Team-Level and Individual-Level Linguistic Styles Affect Patients’ Emotional Well-Being—Evidence from Online Doctor Teams

**DOI:** 10.3390/ijerph20031915

**Published:** 2023-01-20

**Authors:** Xuan Liu, Shuqing Zhou, Xiaotong Chi

**Affiliations:** School of Business, East China University of Science and Technology, Shanghai 200237, China

**Keywords:** online doctor teams, linguistic style, text mining, patients’ emotional well-being

## Abstract

Background: In the post-epidemic era, online medical care is developing rapidly, and online doctor teams are attracting attention as a high-quality online medical service model that can provide more social support for patients. Methods: Using online doctor teams on the Haodf.com platform as the research subject, this study investigates the key factors in the process of doctor–patient communication, which affects patients’ emotional well-being. We also explore the different roles played by doctors as leaders and non-leaders in doctor–patient communication. From the perspective of language style, we select representative factors in the process of doctor–patient communication, namely the richness of health vocabulary, the expression of emotions, and the use of health-related terms (including perceptual words and biological words). We extract both team-level and individual-level linguistic communication styles through textual and sentiment analysis methods and empirically analyze their effects on patients’ emotional well-being using multiple linear regression models. Results: The results show that the expression of positive emotions by the team and attention to patients’ perceptions and biological conditions benefit patients’ emotional well-being. Leaders should focus on the emotional expression, whereas non-leaders should focus on the use of perceptual and biological words. Conclusions: This study expands the application of linguistic styles in the medical field and provides a practical basis for improving patients’ emotional well-being.

## 1. Introduction

The online consultation platform is one of the important products of internet medical development. As a platform bridge between doctors and patients, online consultation platforms can make full use of medical resources [1], provide users with rich health information, and greatly reduce users’ communication costs; they have gradually become an important source for patients to seek social support [2,3]. Doctor–patient communication is the main form of doctor–patient interaction on online consultation platforms, and the ultimate goal of doctor–patient communication is the best health outcome and patient satisfaction [4]. Therefore, enhancing the health outcome of doctor–patient communication and improving patients’ online consultation experience have become important tasks for relevant policymakers and health workers. Health outcomes mainly includes survival, cure, less suffering, pain control, functional ability, vitality, and emotional well-being, within which emotional well-being is the direct result of doctor–patient communication [5]. Emotional well-being refers to the emotional quality of an individual’s everyday experience [6], and dealing with emotions is the most important aspect of doctor–patient communication [7]. In the process of doctor–patient communication, doctors can provide health support or express empathy to reduce patients’ concerns, which may help patients reduce negative emotions and increase positive emotions [8], ultimately improve patients’ emotional well-being. However, how to improve patients’ emotional well-being through doctor–patient communication from a doctor’s perspective remains to be investigated.

Online doctor teams are an emerging service model on online consultation platforms in China [9]. It is a new type of virtual team formed by self-organized doctors to provide patients with multi-to-one online consultation services. In a team, doctors could discuss the medical cases with peer doctors to guarantee the accuracy of diagnosis [10] and provide more comprehensive advices for patients [11]. This arrangement allows doctors’ resources to be reorganized through the internet, which is conducive to giving full play to their professional skills and accomplishing complex medical tasks [12]. Efficient services from doctor teams can enhance patient satisfaction and patient health welfare [13].

Doctor teams usually consist of a leader and other non-leader members. The team leader is the founder of the team [14], and he or she usually plays an important role in teamwork and patient service [15]. According to previous studies, leaders could use their professional capital to organize offline colleagues to build online doctor teams [14]. The team leader’s online performance conveys the image of the team to patients and is an important factor for patients to measure the medical level of the team [9]. In fact, due to their trust in the authoritative expert, many patients will apply for the consultation services of the team led by the experts [16]. Non-leaders are people on the team other than the leader. Non-leaders can choose to join the doctor team that suits them, and they can support the leader to make more comprehensive recommendations for patients. Compared with the individual-level doctor consultation mode, a doctor team provides multi-on-one services and all team members are available to communicate directly with patients when the online consultation begins. Therefore, it is especially important to explore the role of heterogeneous members and collaboratively guarantee the improvement of patients’ emotional well-being in the process of doctor–patient communication.

In online consultation platforms, patients seek support from doctor teams through the internet, and doctor teams interact with patients and supply supports. Text communication is the main form for recent team–patient interaction, thus various supports supplied by doctor teams are usually grounded in online, text-based communications. Linguistic style is a characteristic of the language used by a person in the process of verbal communication [17]. Previous studies have found that in text-based online communication, linguistic style such as emotion, and lexical complexity affect the audience’s perception and judgment of textual information [18]. The linguistic styles of doctor teams and the differences in linguistic style among teams represent, to a certain extent, the unique portrait of the doctor teams. Doctors’ linguistic styles inevitably impact the patients’ emotional well-being. Therefore, it is important to study the influence of linguistic style on the patients’ emotional well-being based on textual information. In addition, each doctor’s response cannot be ignored due to the specificity of the team model. The linguistic styles of various types of team members and their roles in doctor–patient communication outcomes need to be verified.

This study aims to address the following two research questions: (1) Question one: how to quantify the linguistic factors embedded in the interactive texts between doctor teams and patients, and which are the influencing factors that would help to improve patients’ emotional well-being? (2) Question two: Are there different roles for leaders and non-leaders in the online doctor teams in the improvement of patients’ emotional well-being through online interactive communication? This study extends the application of linguistic styles in the doctor team context and explores the underlying mechanism that would benefit patients’ emotional well-being from a novel perspective. The results will help doctor teams use patient-satisfying linguistic styles to improve patients’ emotional well-being, and further provide practical guidance for the subsequent development of online doctor teams.

## 2. Literature Review

An electronic search was undertaken in the Web of Science Core Collection database. To obtain articles related to doctor–patient communication in the online consultation platform, the key words used to perform the research were: TS = (doctor–patient communication) OR TS = (health outcome), in which doctor–patient communication can be replaced by doctor–patient interaction or doctor–patient consultation for retrieval. To obtain articles related to text mining in online consultation platforms, the key words used to perform the research were: TS = (online consultation) AND (TS = (text mining) OR TS = (linguistic styles)).

The remaining articles were screened using the inclusion and exclusion criteria shown in Table 1.

### 2.1. Doctor–Patient Communication and Patient’s Emotional Well-Being

Doctor–patient communication is a multi-channel exchange of all-round information with various characteristics [19], which can meet the informational and emotional health care needs of patients [20,21]. Doctors can improve patient satisfaction by actively communicating with patients in online health platforms [22,23]. Efficient communication facilitates a good, stable, and long-lasting doctor–patient relationship [24], which is an important part of health care practice and is essential for the development of high-quality and efficient health care services [25,26]. In the process of doctor–patient communication, doctors and patients are important participants, with doctors as service providers and patients as service consumers [19]. Patients can search for appropriate medical doctors, describe their symptoms to them, and seek medical advices. The selected doctor then assesses the patient’s health status and provides medical services to the patient [27]. In this process, a large amount of unstructured text information is generated. The quality of the text content of doctor–patient interaction is directly related to patient satisfaction and is the core content of online doctor–patient interaction [19].

The outcomes of doctor–patient communication are another focus on doctor–patient interaction in online consultation platforms. Previous studies have used patient satisfaction [19,28], patients’ doctor selection decisions (whether patients decide to continue the follow-up consultation from a specific doctor) [27] and patients’ emotional well-being [5,29] as consequent variables to measure doctor–patient communication outcomes. Patients’ emotional well-being is widely concerned in the outcome of doctor–patient communication. In the traditional offline consultation scenario, previous literature has also explored the antecedent variables of patients’ emotional well-being improvement. The doctor–patient communication during medical encounters may play a significant role in improving people’s emotional well-being [29]. Communication between doctors and patients is a key pillar of psychosocial support for enhancing the healing process of patients and for increasing their well-being and quality of life [30]. When doctors focus on the emotional needs of their patients, communication outcomes are enhanced [31]. Training in communication skills can help doctors effectively address patient emotions [7] and improve patients’ emotional well-being.

### 2.2. Text Mining in Online Consultation Platform

Unlike the offline consultation channel, within which doctors’ body language and facial expressions can help to transmit information, the current medium of online consultation is mainly text-based interaction between physicians and patients. Therefore, previous researchers have focused to investigate the interactive content of online consultation platforms using “text mining” methods.

“Text mining” refers to the use of natural language processing (NLP) techniques to extract facts, relationships, and opinions from a text [32]. With the rise of social media, massive amounts of text data have emerged on the web, and the study of social media content based on text mining has received attention from scholars. The methods of text analysis are applied in various fields, such as online risk assessment [33], user consumption [34,35], business management [36], and health care [37,38].

Patients participate in online consultation platforms by seeking and providing content in online forums [39]. Many scholars have applied text-mining methods to extract informational and emotional exchanges in online health communities [40], and emotional support interactions among users are far more frequent than information support exchanges [41]. Table 2 shows the application of text mining in online consultation platforms. Most studies analyze user texts from either informational and emotional dimensions, and most studies focus on patient-to-patient communication rather than doctor-to-patient communication.

In addition, text mining is also an important tool to extract linguistic style. Linguistic style refers to the habits of a person’s language usage. According to the linguistic analysis theory of natural language processing, linguistic style includes five aspects: morphological, lexical, syntactic, semantic, and pragmatic [45]. Because morphology refers to the use of morphemes, such as roots and suffixes in English and radicals in Chinese, which have extremely small effects on communication effects, previous studies mainly focus on following four dimensions: lexical, syntactic, semantic, and pragmatic. Table 3 shows the meaning of each dimension and the represented variables.

### 2.3. Research Gaps

Review of previous studies reveals the following two main aspects:Previous studies have recognized patients’ emotions as an important issue in doctor–patient communication, but limited studies have used secondary data to explore patients’ emotional health in doctor–patient communication through empirical methods, especially at the team level.Language is the basic form of doctor–patient communication in online consultation platform, so doctors’ linguistic styles may affect the patients’ perceptions and judgments of the communication process. However, few studies have unfolded the textual interactions produced by doctor teams from the perspective of linguistic style.

In this study, we applied the research framework of linguistic style to the textual interactions of online doctor teams and patients, analyzed the overall linguistic style of the teams, and explored the influence of linguistic style on the patients’ emotional well-being. We also investigate the different roles of team leaders and non-leaders when participating in team consultations.

## 3. Hypothesis Development

This study examines the linguistic style of online doctor teams and its role in the doctor–patient interaction, both in teams and on an individual level. On the team level we explore the impact of overall team communication behavior on patients’ emotional well-being. On the individual level we further explore the influence of two different roles, leader and non-leader members, on the performance of doctor–patient communication. According to linguistic analysis theory, the linguistic style dimensions affecting communication outcomes include four aspects (lexical, syntactic, semantic, and pragmatic). In the doctor–patient communication scenario, this study explored the impact of the doctor’s linguistic style on the emotional well-being of the patient from both lexical (vocabulary richness) and semantic (health-related terms and sentiment) aspects.

### 3.1. Vocabulary Richness and Patients’ Emotional Well-Being

On online consultation platforms, the registered doctors may differ in the level of vocabulary richness they use to provide services because of their diverse professional backgrounds, education levels, and work experiences. Compared to other linguistic features, it is easier for a person to control the vocabulary richness used for expression [17]. People with a rich vocabulary can express themselves better and thus influence others [46]. It has been shown that in online knowledge communities, the use of richer vocabulary by users during communication can engage more of an audience [48]. In the health care domain, studies based on language styles have shown that the patients’ vocabulary richness in health-related terms also significantly affects their chances of receiving information support [17]. As for doctors, we infer that doctors who are familiar with the names of various diseases, symptoms, drugs, treatment methods, and body parts may express their recommendations effectively, thus enhance the efficiency of communication and the outcome of consultation services. At the same time, the doctor’s use of rich health vocabulary can also demonstrate the doctor’s professionalism and increase the patient’s confidence in the cure of the disease. Therefore, it is expected that vocabulary richness has a positive effect on patients’ emotional well-being when online doctor teams consult with patients. To verify the role of vocabulary richness during online consultations with patients, the following hypotheses are proposed from the team level and the individual level:

**H1a:** Vocabulary richness of the online doctor teams benefits patients’ emotional well-being.

**H1b:** Vocabulary richness of leaders in online doctor teams benefits patients’ emotional well-being.

**H1c:** Vocabulary richness of non-leader members in online doctor teams benefits patients’ emotional well-being.

### 3.2. Health-Related Terms and Patients’ Emotional Well-Being

The use of health-related terms is another important linguistic style factor in online consultation platform due to the specialized nature of the interactions that occur on online health care platforms. Studies have found that medical experts and patients differ in their understanding of the same medical terminology [49,50,51,52], so a more generic health language may provide more accurate and effective help to patients. For different types of diseases, common health-related terms during the consultation are reflected in both perceptual words and biological words. Perceptual words focused on the patient’s perceptions, including see-, hear-, and feel-related words. Biological words focus on physiological processes, including words that relate to the body, health, sexuality, and ingestion (see more details in Table A1). At the team level, we expected teams, as a whole, which pay more attention to patients’ perceptions, symptoms, and hygiene habits could help patients understand their physical conditions, thus benefiting patients’ emotional well-being. At the doctor level, we proposed that both leaders and non-leaders with a high level of both health-related terms could play a positive role. We propose the following hypotheses:

**H2a:** The use of perceptual words by the online doctor teams benefits patients’ emotional well-being.

**H2b:** The use of perceptual words by leaders in online doctor teams benefits patients’ emotional well-being.

**H2c:** The use of perceptual words by non-leader members of the online doctor teams benefits patients’ emotional well-being.

**H3a:** The use of biological words by the online doctor teams benefits patients’ emotional well-being.

**H3b:** The use of biological words by leaders in online doctor teams benefits patients’ emotional well-being.

**H3c:** The use of biological words by non-leader members of the online doctor teams benefits patients’ emotional well-being.

### 3.3. Emotional Expression and Patients’ Emotional Well-Being

In the health consultation platform, emotional support interactions are more frequent than information support exchanges [41], and emotional support may contribute to the treatment of disease [30,43]. At the team level, the overall transmission of positive emotions facilitates a more relaxed emotional state of the patient. Positive emotions from multiple members can enhance patient trust and may play a positive role in the improvement of patients’ emotional well-being. At the individual level, positive emotions of both leaders and non-leaders lead patients to become more emotionally positive, which will have a positive effect on their emotional well-being. Thus, the following hypotheses are proposed:

**H4a:** Positive emotional expression of the teams in online doctor teams benefits patients’ emotional well-being.

**H4b:** Positive emotional expression of the leaders in the online doctor teams benefits patients’ emotional well-being.

**H4c:** Positive emotional expression of the non-leaders in the online doctor teams benefits patients’ emotional well-being.

The conceptual model of this study is summarized in Figure 1.

## 4. Methodology

### 4.1. Sample and Data Collection

The doctor team, as a new medical service model on online consultation platforms in China, has been emerging since 2017 and is now widely used. From the data we obtained, as of 2020, there are over 2000 doctor teams serving 182,000 patients on the leading online consultation platform—Haodf online platform (www.Haodf.com (accessed on 15 January 2023)), in China. This shows that the doctor team model has gained universal recognition. The data for this study were obtained from the online consultation platform (www.Haodf.com (accessed on 15 January 2023)), and the consultation data of doctor teams from June 2017 to November 2019 were crawled. In addition, the individual data of doctor members in doctor teams and the team characteristics were crawled separately, among which the individual data of doctors included the doctor’s title, department, hospital, and the like, and the characteristics of doctor teams included the team size, team price, team composition, number of patients served by teams, team response rate, team establishment duration, and so on. Our dataset contains a total of 1318 doctor teams and 871,399 interactive texts. A sample homepage of the doctor team is shown in Figure 2. As we can see from the figure, doctor teams are generally named using the leader’s name, such as “Professor Liver Cancer Diagnosis and Treatment Team,” which indicates that the team is led by Professor and the team focuses on liver cancer diagnosis and treatment. The page also lists doctors within the team, including the team leader and all non-leader members, allowing patients to easily know the composition of the focal team. The leader appears first in the doctor list and is labeled as the team leader, making it easy for patients to distinguish between team leaders and non-leaders. A sample consultation page is shown in Figure 3. All team members are available to communicate with patients, but user-related information (such as pictures, etc.) is not displayed during the consultation process in order to protect user privacy. Based on the doctor–patient interactive texts, from the team and the individual member level, the linguistic characteristics of online doctor teams were calculated at the communication level to explore the factors that affect patients’ emotional well-being. All data used in this study is non-private.

### 4.2. Variable Measurement

#### 4.2.1. Dependent Variable

Patients’ Emotional Well-Being: Patients’ emotional well-being measures the direction and extent of the patient’s emotional change during the consultation with the doctor’s team. Patients’ emotional well-being was calculated in five steps. In the first step, sessions were sliced (i.e., multiple sessions between the same patients and doctors were split into single consultation sessions according to the start mark of each consultation). In the second step, valid sessions were screened; 15% of the sessions that were too long or too short were deleted. After that, based on the number of remaining text samples, we identified the middle position of the patient’s inquiry texts in each consultation and further confirmed whether it occurred after the doctor’s reply (to make sure the patient’s emotional well-being is due to the doctors’ effective communications). In this way, we obtain a valid conversation. The third step involved calculating the sentiment value; after cutting the patient’s texts in the valid session into two parts, we used the Chinese sentiment analysis tool SnowNLP to calculate the sentiment value of the two parts (see Appendix B). In the fourth step, patients’ emotional well-being was calculated by subtracting the emotional value of the first part from the emotional value of the second part to calculate the positive changes in the patient’s emotions for each consultation. A positive value indicates that the communication with the doctor team improved the patient’s overall emotions, while a negative value indicates that the communication with the doctor team negatively affected the patient’s overall emotions. Finally, the changes in patients’ emotions were aggregated to the team level, and mean values [44,53] were taken to indicate the impact of online doctor teams on patients’ emotional well-being in the general process of communication. Higher values of patients’ emotional well-being indicate that the information and emotional support given by the doctor teams during the consultation effectively guided or calmed the patient’s emotions, by reducing patients’ negative emotions or increasing their positive emotions.

#### 4.2.2. Independent Variable

Vocabulary Richness: Vocabulary richness, in the general context, are usually aimed at assessing users’ vocabulary richness in general health language. Since our study is a health consultation scenario, this study emphasized the usage of perceptual words and biological words from LIWC as a health lexicon (see Appendix A). First, all the doctor–patient interactive texts in the sample were divided into words, and the words in the categories of perceptual words and biological words in LIWC 2017 were filtered out using TextMind. Then the number of de-weighted health words in each interactive text (i.e., the number of unique health words) was calculated. Since the number of unique health words increases as the total number of words increases, to reduce the effect of the total number of words on the health word richness measure, the Dugast formula [54] was used to calculate the vocabulary richness for a team (Team Richness), leaders (Leader Richness), and non-leader members (Non-leader Richness):(1)$$D=\frac{{\left(\log(total\;word\;count)\right)}^{2}}{\log(total\;word\;count)-\log\left(unique\;health\;words\right)}$$

Health-Related Terms Use (Perception and Bio): In this study, the software TextMind was used to calculate the health-related terms for the interactive texts (see Appendix A). For each interactive text, the number of perceptual words and biological words was first identified separately as a percentage of the total text content. The mean values of the corresponding variables at the team level [44,53] were used to calculate the health-related terms use of all team members (Team Perception, Team Bio), leaders (Leader Perception, Leader Bio), and non-leaders (Non-Leader perception, Non-Leader bio). In contrast to health vocabulary richness, repeated words were counted multiple times in the calculation of health-related terms use.

Emotional Expression: In this study, we used SnowNLP (see Appendix B) to calculate the overall emotional expression of teams (Team Emotion), team leaders (Leader Emotion), and non-leader members (Non-Leader Emotion) in the process of doctor–patient communications. The larger value indicated higher positive sentiment.

#### 4.2.3. Control Variables

Various control variables were extracted at the team level. Team Prices indicates the quality of the team’s service. A team with a higher quality of service is more likely to satisfy patients, and the communication process between doctors and patients is more effective. Team Longevity may affect the quality of communication and collaboration between team members. Team Size indicates the number of team members, and the effective collaboration of members in a larger team can highlight the advantages of team services on a multi-to-one basis. Reply Rate indicates the efficiency of the team. Team Help characterizes the commitment of all team members on the platform. Team Comprehensive Level represents the overall service level of the team members. In addition, the Leader Involvement Ratio may have an impact on the quality of team communication and collaboration. Disease Seriousness is added to control the difficulty of disease treatment, which may affect the communication between the doctor team and the patient. Patient Health Terms calculate the patient’s health literacy, which may affect the patient’s understanding of the doctor–patient communication process. The variable types and variable names are shown in Table 4.

#### 4.2.4. Regression Models

The study selected a multiple linear regression model to conduct empirical analysis. Multiple linear regression allows fitting the relationship between multiple independent variables and dependent variables, which is widely used in studies related to online consultation platforms [38,40,55]. Patients’ emotional well-being at the team level and individual level are shown in Equations (2) and (3).
(2)$$\begin{array}{c} {Y_{Patients^{’}EmotionalWell-Being-team}}_{i}\\ =\cdot \beta_{0}+\beta_{1}*TeamRichness_{i}+\beta_{2}*TeamPerception_{i}+\cdot \beta_{3}*TeamBio_{i}+\beta_{4}\\ {}*TeamEmotion_{i}+\beta_{5}*TeamPrice_{i}+\beta_{6}*TeamLongevity_{i}+\cdot \beta_{7}*Teami+\beta_{8}\\ {}*ReplyRate_{i}+\beta_{9}*TeamHelp_{i}+\beta_{10}*TeamComprehensiveLevel_{i}+\cdot \beta_{11}\\ {}*LeaderInvolvementRatio_{i}+\beta_{12}*DiseaseSeriousness_{i}+\beta_{13}PatientHealthTerms_{i}+\delta_{i} \end{array}$$
(3)$$\begin{array}{c} {Y_{Patients^{’}EmotionalWell-Being-member}}_{i}\\ =\cdot {\beta_{0}}^{\prime }+{\beta_{1}}^{\prime }*LeaderRichness_{i}+{\beta_{2}}^{\prime }*LeaderPerception_{i}+{\beta_{3}}^{\prime }*LeaderBio_{i}\cdot +{\beta_{4}}^{\prime }\\ {}*LeaderEmotion_{i}+{\beta_{5}}^{\prime }*NonleaderRichness_{i}+{\beta_{6}}^{\prime }*NonleaderPerception_{i}\cdot +{\beta_{7}}^{\prime }\\ {}*NonleaderBio_{i}+{\beta_{8}}^{\prime }*NonleaderEmotion_{i}+{\beta_{9}}^{\prime }*TeamPrice_{i}\cdot +{\beta_{10}}^{\prime }\\ {}*TeamLongevity_{i}+{\beta_{11}}^{\prime }*Teami+{\beta_{12}}^{\prime }*ReplyRate_{i}+\cdot {\beta_{13}}^{\prime }*TeamHelp_{i}+{\beta_{14}}^{\prime }\\ {}*TeamComprehensiveLevel_{i}\cdot +{\beta_{15}}^{\prime }*LeaderInvolvementRatio_{i}+{\beta_{16}}^{\prime }\\ {}*DiseaseSeriousness_{i}+{\beta_{17}}^{\prime }PatientHealthTerms_{i}+{\delta_{i}}^{\prime } \end{array}$$

## 5. Results

### 5.1. Descriptive Statistical Analysis

Table 5 demonstrates the descriptive statistical analysis of all variables, including the maximum, minimum, mean, and standard deviation of the variables. The mean value of the dependent variable Patients’ Emotional Well-Being is 0.017, indicating that the patient emotion is, on average, seen to increase in a positive direction after communication with the doctor team, and the overall communication between the doctor team and the patient within the session is effective.

### 5.2. Correlation Analysis

The correlations between the independent variables and the control variables at the team level are shown in groupings in Table 6, and the correlation coefficients between the variables are all less than 0.6, without excessive values, indicating the absence of multicollinearity, and are suitable for further analysis.

### 5.3. Regression Analysis

Table 7 shows the regression results. Model 1 adds control variables. Model 2 adds team-level independent variables based on Model 1 to explore the research question one: The impact of team’s linguistic style on patients’ emotional well-being (H1a, H2a, H3a and H4a). Model 3 adds individual-level independent variables based on Model 1 to explore Question two: The impact of the linguistic style of leader and non-leader members within the team on patients’ emotional well-being (Leader level includes H1b, H2b, H3b and H4b; No-leader level includes H1c, H2c, H3c and H4c).

Models 1 to 3 are used to examine the effect of doctors’ linguistic styles on patients’ emotional well-being. Only control variables are included in model 1. The results show that Team Price (β = 0.009, *p* = 0.048) is positively associated with patients’ emotional well-being. This may because team price is a signal of service quality, and high service quality may increase patients’ emotional well-being. Team Longevity (β = −0.037, *p* = 0.027) and Disease Seriousness (β = −0.034, *p* = 0.025) are negatively correlated with patients’ emotional well-being. A team that has been registered for a long time may decrease their team activities, and the quality of service to patients may decline. Patients with mild diseases are more likely to be positively influenced by doctor–patient communication, and thus have their emotional well-being improved.

Model 2 adds team-level independent variables to model 1, including Team Richness, Team Perception, Team Bio, and Team Emotion, to test hypotheses H1a, H2a, H3a, and H4a. Model 2 shows that the effect of vocabulary richness (β = 0.001, *p* = 0.839) on patients’ emotional well-being is not significant, indicating that H1a is not supported. The vocabulary richness in the doctor’s response is not important; this may illustrate that doctors should focus on patients’ specific concerns rather than on using distracting and inconsistent vocabulary. Model 2 also shows the team’s use of both perceptual words (β = 1.498, *p* = 0.001) and biological words (β = 0.488, *p* = 0.001) benefit patients’ emotional well-being improvement, indicating that H2a and H3a are both supported. Thus, in the process of doctor–patient communication, doctors should pay more attention to the patient’s perception and biological condition, which will make the patient feel that their problem is being solved and their anxiety may be relieved. In terms of emotional expression, the more positive the team’s emotions are, the higher the patients’ emotional well-being will be (β = 0.170, *p* < 0.001), and H4a is supported. One possibility is that the positive emotions of the team signal to the patient that the disease is easily cured, or that the patient’s emotions are boosted by the positive emotions of the doctor’s team. Compared to model 1, control variable Patient Health Terms in model 2 became significant, while maintaining a negative effect. A possible reason is that patients with higher health literacy would have a clearer understanding of the disease, when communicating with doctors; their negative emotions would hardly be relieved and their positive emotions would be improved less obviously.

Model 3 added individual-level independent variables to model 1, including the Leader Richness (H1b), Leader Perception (H2b), Leader Bio (H3b), Leader Emotion (H4b), Non-Leader Richness (H1c), Non-Leader Perception (H2c), Non-Leader Bio (H3c), and Non-Leader Emotion (H4c). Consistent with Model 2, vocabulary richness on the individual level also shows no benefit to patients’ emotional well-being. Both H1b and H1c are not supported. In the usage of health-related terms, the use of perceived words (β = 0.585, *p* = 0.101) and biological words (β = 0.069, *p* = 0.503) by leaders have no significant effect on patients’ emotional well-being, indicating that H2b and H3b are not supported. However, the use of perceived words (β = 0.671, *p* = 0.031) and biological words (β = 0.358, *p* = 0.001) by non-leaders is beneficial to patients’ emotional well-being, showing that both H2c and H3c are supported. The results indicate that the use of health-related terms by non-leader members rather than leaders played the major role in affecting patients’ emotional well-being. In terms of emotional expression, the leader’s emotions would drive patients’ emotions in the same direction (β = 0.137, *p* < 0.001), while the non-leader members’ positive emotion expression had a negative effect on patients’ emotional well-being (β = −0.421, *p* < 0.001). The results support H4b but do not support H4c. One possible reason is that the positive emotions of the leader are more persuasive due to the authority of the expert, and the encouragement and sympathy from leaders will have a greater impact on guiding the positive emotions of patients and relieving their negative emotions; In contrast, the positive emotions of the non-leader members may cause patients to feel worthless, because they prefer non-leaders to focus on their health problems and care about their biological or perceptual status. The results may suggest that patients would trust a leader much more than non-leader members in the doctor teams, and they prefer to receive positive feedback from an authority figure.

Comparing models 2 and 3, the following conclusions could be drawn. For vocabulary richness, the results reveal that the effect of vocabulary richness on patients’ emotional well-being at both the team level and the individual level is not significant, which sufficiently shows that, in doctor–patient communication, teams or doctors do not need to use overly rich vocabulary. Teams and doctors should focus on patient concerns, since using overly rich vocabulary may make it more difficult for patients to understand. For health-related terms, the use of perceived words and biological words by team and non-leaders have a positive impact on patients’ emotional well-being, whereas the use of perceived words and biological words by leaders have positive but insignificant effect on patients’ emotional well-being. The results suggest that non-leaders should pay more attention to the patient’s health status during the doctor–patient communication process, and make suggestions that address patients’ concerns. For emotional expression, patients are more optimistic and emotionally relaxed when the team as a whole showed positive emotions. However, at the individual level, positive emotions of leaders positively affect patients’ emotional health, while non-leaders have a negative impact. The results indicate that patients prefer to receiving encouragement and emotional support from the leaders within doctor teams, but have some resistance to the emotional expression from non-leaders.

### 5.4. Discussion

Our research has the following theoretical contributions: (1) This study quantifies the doctor–patient communication process using conceptions in linguistic styles, and explores the underlying mechanisms of patients’ emotional well-being. In our study, we incorporate linguistic style theory and explore the effects of three linguistic concepts from lexical and semantic aspects in linguistic style dimensions, namely vocabulary richness, use of health-related terms, and emotional expressions, on the doctor–patient communication outcomes. Previous studies on doctor–patient communication have shown that doctor–patient communication could support information exchange by and benefit from responding to emotions [5]. Both informational and emotional support from doctors positively affects patient satisfaction [19]. This study further used text mining and sentiment analysis methods to quantify the doctor–patient communication process; to be more specific, we extracted the linguistic styles using by doctor teams grounded in textual communications with patients, and emphasize the patient’s emotional well-being to measure the direct outcome of doctor–patient communication. The results reveals that the use of health-related terms and positive emotional expression by doctor teams are beneficial to the patients’ emotional well-being. The findings of the study enrich the application of text mining and sentiment analysis methods in the online health consultation field, as well provide new perspectives for a comprehensive understanding of the relationship between doctors’ linguistic styles and patients’ emotional well-being. In the meanwhile, our research also contributes to the extant literature on information exchange and emotional exchange between doctors and patients. (2) Unlike those studies which regard the doctor team as a whole, our study unfolds the team into leader and non-leader members, and explored the effects of different roles on the outcomes of doctor–patient communication. The main findings of the study include: compared to non-leaders, positive emotional expressions by leaders are more important to enhance patients’ emotional health, and the use of health-related terms by non-leaders contributes more to patients’ emotional well-being. Although previous articles have recognized that members within a team may have different roles and divisions of labor [56,57], most of them have focused on team leaders, such as identifying that leaders play a key role in regulating team behavior [58,59] and managing instability within teams [58,60]. This paper explores the heterogeneous roles of different members in the process of doctor–patient communication, and provides a new way to open the black box of the doctor team consultation process. The findings could enrich the research in the field of online doctor teams and provide new ideas for further research on communication and collaboration within doctor team.

The research findings also have practical implications: (1) For doctor teams, this study is useful in guiding online doctor teams on how to adapt language expressions during doctor–patient communication so as to enhance the health outcomes. On the one hand, the doctor team should focus on the use of health-related terms and pay close attention to the health needs of patients. The use of perceptual words such as “gaze”, “listen”, and “touch” can help patients pay more attention to their personal conditions; the use of biological words such as “medicine”, “cold,” and “ gut” can also help patients pay attention to their disease symptoms and follow doctors’ medical advices. On the other hand, doctor teams should also properly use emotional expression when providing consulting services. Positive emotions could motivate patients and relieve their anxiety about the disease, thus further enhancing patients’ emotional well-being. (2) For members in different roles within the team, the findings of this paper suggest that both leaders and non-leaders play critical roles in the patients’ emotional well-being, however, heterogeneous members play their roles differently. Therefore, when providing consultation services, team participants in different roles should consider using appropriate communication styles to better serve to patients. For leaders, the expression of positive emotions is even more important than the use of health-related terms. Positive emotions can demonstrate their confidence in the treatment of the disease, thus driving the patient’s emotions to improve. Non-leader members should collaborate with the leader to pay more attention to the patient’s health needs and use more generalized health language in communication process. At the same time, it is important for non-leaders to reduce the expression of emotions and focus the patient’s attention on the consultation related to the disease. (3) For online consultation platforms, since the process of doctor–patient communication has a direct impact on patients’ emotional well-being, the platforms should further think about how to better guide doctors and patients to communicate in an appropriate way. On the one hand, the platform can consider providing communication guidelines for doctors and patients. Effective communication guidelines can not only help patients better express their health conditions, but also help doctors adjust their linguistic style according to patients’ health literacy. On the other hand, the platform can use IT to assist doctor–patient communication. For example, IT can be used to identify key medical terms in the process of doctor-to-patient communication. The platform can generate a list of doctor’s suggestions and health tips for patients, so as to help patients better follow the doctors’ medical advices for treatment. IT can also be used to recommend a doctor team based on the symptoms described by the patient. The diagnostic opinions of similar cases in the past can be analyzed and used by the platform to recommend the suitable doctor team and help patients to make quick inquiries.

## 6. Conclusions

The achievement of complex team tasks requires contributions from each type of member, including leaders and non-leaders. This study innovatively investigates factors that affect online doctor team and patient communication from a linguistic style perspective, and further explores how the linguistic styles of leaders and non-leaders within doctor teams play different roles in improving the patients’ emotional well-being. Based on the texts of doctor–patient interactions in online consultation platforms, the research used text mining and sentiment analysis methods to extract the relevant influencing factors in terms of vocabulary richness, use of perception related vocabulary, use of biological related vocabulary, and emotional expression, and empirical analysis at the team level and individual level are explored using multiple linear regression. Two main conclusions can be drawn: (1) For question one, the linguistic style used by the online doctor team affects the patients’ emotional well-being, specifically, the positive emotional expression and attention to the patient’s perceived and biological condition by teams help to improve patients’ emotional well-being; (2) For question two, improving patients’ emotional well-being requires each type of team member to use health vocabulary rationally and express emotion appropriately. Patients’ positive emotions are more likely to be driven by leaders’ positive emotions, and as an authoritative expert, leaders should give positive signals to patients as much as possible based on solid consultation decisions, which will help patients to build up an optimistic attitude towards the disease. Non-leader members should rather pay more attention to the use of health-related terms in the interactive communication process, which is conducive to patients’ emotional well-being.

The current study also has the following limitations: (1) Our research is an empirical study based on online doctor teams, an emerging service model that is now widely used in online consultation platforms in China. However, the applicability of the model in other countries remains to be further explored, and the generalization ability of underline mechanisms discovered in this study needs to be further investigated. (2) Doctor–patient communication is a two-way interactive process. In our model, we only considered patients’ use of health-related terms as a patient-related variable to measure patients’ health literacy. In addition, due to privacy problems, the study cannot examine the effect of other patient characteristics (e.g., patient age or gender) on the outcome of doctor–patient communication. In the future, when data are further enriched, more factors can be included in the model. (3) We examined the impact of different roles of physicians on the outcomes of doctor–patient communication. However, we do not consider the possible conflicts between doctors of different roles in providing consultation services. Future research could further consider the collaboration and conflict among team members.

## Figures and Tables

**Figure 1 ijerph-20-01915-f001:**
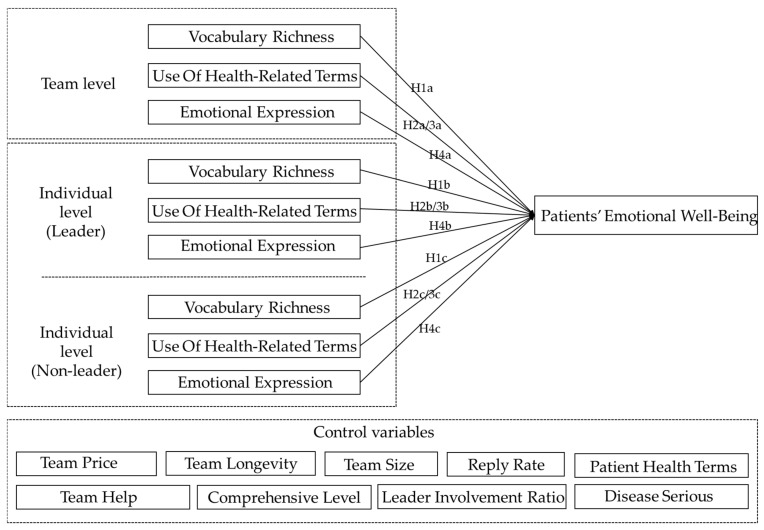
Research model on the influence mechanism of patients’ emotional well-being in online doctor teams.

**Figure 2 ijerph-20-01915-f002:**
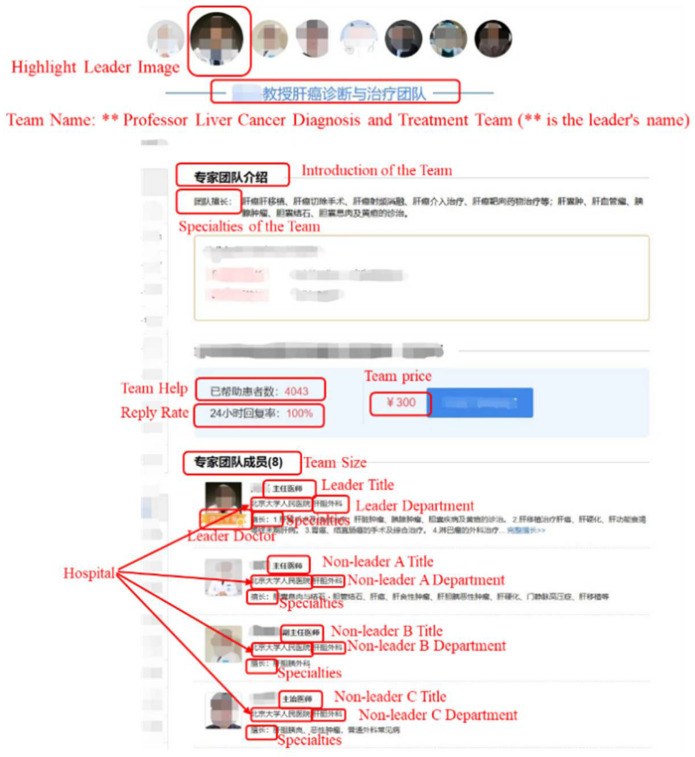
Homepage of online doctor team.

**Figure 3 ijerph-20-01915-f003:**
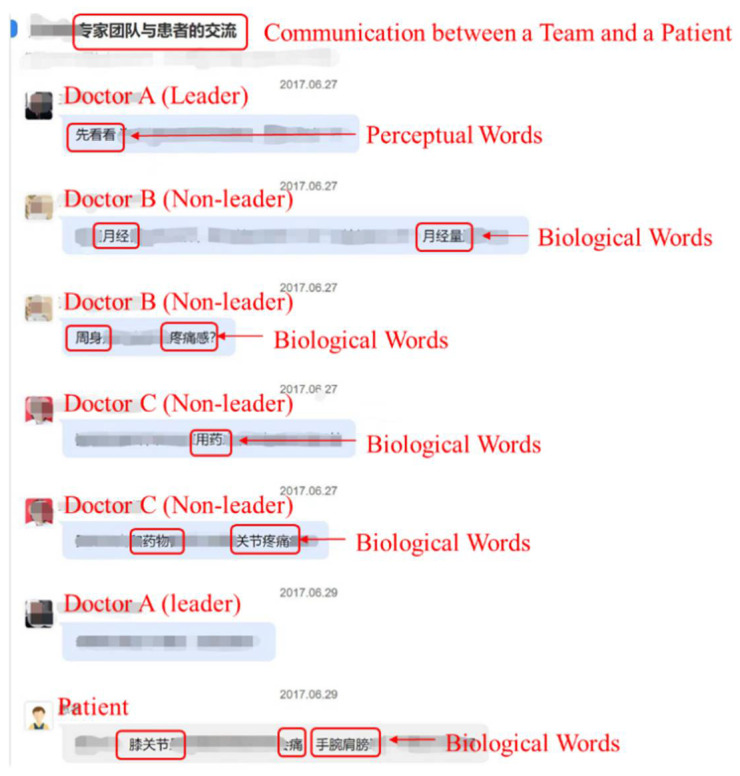
Doctor–patient communication page with patients (incomplete screenshot).

**Table 1 ijerph-20-01915-t001:** Inclusion and exclusion criteria.

Inclusion Criteria	Exclusion Criteria
Empirical Research	Non-English studies
Review Article	Conference Abstracts
Health outcomes of doctor–patient communication	Not pertinent to the field of investigation
Information and emotional support in online consultation platform	

**Table 2 ijerph-20-01915-t002:** Application of text mining in online consultation platform.

Dimensionality	Author	Interaction Mode/Scenario	Research Content
Information	Peng et al. [40]	doctor-to-patient information disclosure	Aims: identify potential topics in doctors’ self-disclosure information, and explore the impact of topic diversity in doctor self-disclosure on patient choice. Methods: LDA topic model; hierarchical clustering method Results: excessive quantity of information and semantic topic diversity can raise barriers for patient’s decision.
	Park et al. [42]	patient-to-patient communication	Aims: examine how different types of supportive messages posted on OHCs encourage users to increase their health resilience. Method: directed content analysis Results: self-efficacy-oriented messages affect helpfulness, while response-efficacy-oriented messages influence the relationships among helpfulness, goal-setting, and health resilience.
Emotion	Lu et al. [43]	patient-to-patient communication	Aims: calculate the emotional representation of depressed patients in texts from an online consultation platform, and further investigate whether the use of online communities helps improve depression. Methods: Baidu AI’s natural language processing method Results: Emotional support positively affect the treatment of depression.
	Liu et al. [37]	patient-to-patient communication	Aims: explore various patterns of information exchange and social support in web-based health care communities and identify factors that affect such patterns. Methods: social network analysis; text mining techniques Results: polarized sentiment increases the chances of users to receive replies, and optimistic users play an important role in providing social support to the entire community.
Information and emotion	Chen et al. [44]	patient-to-patient communication	Aims: consider whether or not linguistic signals in posts (including sentiment valence, linguistic style matching, readability, post length, and spelling) impact the amount of support received. Methods: social support classification using SVM; structured information extraction Results: affective linguistic signals, including negative sentiment and linguistic style matching, are effective in invoking both informational and emotional support from the community.
	Jiang et al. [17]	patient-to-doctor communication	Aims: how various linguistic characteristics of patients’ communication in these communities affect their social support outcomes. Methods: linguistic analysis; exponential random graph models Results: lexical richness in health-related vocabulary negatively correlates with receiving informational support. The readability and brevity of written texts have positive relationships with incoming social support.

**Table 3 ijerph-20-01915-t003:** Language style dimensions and variables.

Dimensionality	Perspective	Representative Variables
Lexical	the usage of characters and words in sentences	Vocabulary richness [46] (which measures how many different words one uses in communication, and the use of richer words tends to be more persuasive)
Syntactic	grammar and the appropriate use of words in sentences	Readability of information [17] (refers to the extent to which the content can be easily understood by an intended audience) The length of information [44] (in general, longer texts can convey richer information, and text length can positively affect the amount of information received)
Semantic	the meanings of words behind their occurrence	Sentiment [39,40] (doctors who show more emotional support may enhance patient satisfaction and trust in the doctor teams) Content-specific keywords [17] (patients can determine to what extent they discuss topics related to diseases, symptoms, treatments, and their relations)
Pragmatic	the meaning of words and word choice in the appropriate contexts; Distinct from semantics, pragmatics concerns the choice of words used to express the same meaning	Level of language sharing [44,47] (higher levels of language sharing in a given scenario can increase communication effectiveness [44] and willingness to share knowledge [47]. Using health language that other OHC participants also use may increase one’s social acceptance and lead to better communication outcomes [17])

**Table 4 ijerph-20-01915-t004:** Variable definition table.

Variable Type	Variable Name	Variable Definition
Dependent variable	Patients’ Emotional Well-Being	The direction and extent of the patient’s emotional change during the consultation with the doctor’s team
Independent variable —— Team level	Team Emotion	The emotion of the doctors’ reply text, range from 0 (negative) to 1 (positive)
	Team Perception	The percentage of perceptual words in team level usage
	Team Bio	The percentage of biological words in team level usage
	Team Richness	Health vocabulary richness in team level
Independent variable —— Individual-Leader level	Leader Emotion	The emotion of the leader’s reply text, range from 0 (negative) to 1 (positive)
	Leader Perception	The percentage of perceived words in leader’s conversations
	Leader Bio	The percentage of biological words in leader’s conversations
	Leader Richness	Health vocabulary richness in leader’s conversations
Independent variable —— Individual-Non-Leader level	Non-Leader Emotion	The emotion of the non-leader’s reply text, range from 0 (negative) to 1 (positive)
	Non-Leader Perception	The percentage of perceived words in non-leader’s conversations
	Non-Leader Bio	The percentage of biological words in non-leader’s conversations
	Non-Leader Richness	Health vocabulary richness in non-leader’s conversations
Control variables	Team Price	The natural logarithm of the team price of an online doctor team
	Team Longevity	As measured by the number of days between team inception and the deadline for data collection (log-transformed)
	Team Size	The number of team members
	Reply Rate	The response rate of a team answered within 24 h from a patient’s question being asked
	Team Help	The total number of patients served by the team on the online medical consultation platform since its inception (log-transformed)
	Team Comprehensive Level	Obtained by taking the mean values of the title level, the hospital level, and the city level of the team members, respectively, and standardizing them, and then summing them
	Leader Involvement Ratio	The ratio of the number of consultations involving team leaders to the total number of team consultations
	Disease Seriousness	The difficulty of treatment of the disease, serious diseases such as cancer, leukemia, uremia, AIDS, and heart disease were set to 1, and other diseases were set to 0
	Patient Health Terms	The use of health-related terms by patients. For each interaction text of the patient, the number of perceptual and biological words as a percentage of the total text content was calculated using TextMind (see Appendix A) and then aggregated to the team level.

**Table 5 ijerph-20-01915-t005:** Variable descriptive statistics (n = 1318).

Variable	Max	Min	Average	Standard
Patient’s Emotional Well-Being	0.885	−0.937	0.017	0.145
Team Richness	10.577	1.176	6.486	1.596
Team Perception	0.126	0.000	0.013	0.009
Team Bio	0.272	0.000	0.094	0.030
Team Emotion	0.988	0.027	0.567	0.103
Leader Richness	10.297	0.778	5.535	1.707
Leader Perception	0.134	0.000	0.013	0.011
Leader Bio	0.444	0.000	0.094	0.039
Leader Emotion	0.999	0.002	0.563	0.124
Non-Leader Richness	9.736	0.954	5.790	1.737
Non-Leader Perception	0.176	0.000	0.015	0.013
Non-Leader Bio	0.400	0.000	0.095	0.037
Non-Leader Emotion	0.964	0.021	0.543	0.086
Team Price	6.868	1.792	3.851	0.965
Team Longevity	6.813	5.153	6.397	0.268
Team Size	10.000	2.000	3.519	1.509
Reply Rate	1.000	0.000	0.873	0.218
Patient Health Terms	0.542	0.000	0.119	0.037
Leader Involvement Ratio	1.000	0.000	0.652	0.360
Disease Seriousness	1.000	0.000	0.218	0.274
Team Comprehensive Level	1.359	−4.059	0.004	0.588
Team Help	6.695	0.693	3.134	1.215

**Table 6 ijerph-20-01915-t006:** Table of the correlation coefficient (n = 1318).

	1	2	3	4	5	6	7	8	9	10	11	12	13
1. Team Price	1												
2. Team Longevity	0.206 ***	1											
3. Team Size	0.118 ***	0.211 ***	1										
4. Reply Rate	0.065 **	0.00900	−0.0140	1									
5. Team Help	0.252 ***	0.401 ***	0.239 ***	0.086 ***	1								
6. Team Comprehensive Level	0.068 **	0.0360	−0.086 ***	−0.049 *	0.0150	1							
7. Leader Involvement Ratio	0.144 ***	0.051 *	−0.093 ***	0.055 **	−0.0220	−0.076 ***	1						
8. Disease Seriousness	0.099 ***	−0.074 ***	0.097 ***	−0.078 ***	−0.052 *	0.088 ***	−0.046 *	1					
9. Patient Health Terms	−0.159 ***	−0.0350	−0.0350	−0.0140	−0.104 ***	−0.071 **	−0.073 ***	−0.067 **	1				
10. Team Richness	0.138 ***	0.261 ***	0.156 ***	0.051 *	0.512 ***	−0.00600	−0.0130	−0.064 **	−0.0190	1			
11. Team Perception	−0.052 *	−0.063 **	−0.0410	0.00400	−0.00900	−0.049 *	0.0170	−0.122 ***	0.089 ***	0.055 **	1		
12. Team Bio	−0.092 ***	−0.00700	0.0260	−0.0180	−0.103 ***	−0.00700	−0.0310	0.0140	0.379 ***	0.00800	−0.049 *	1	
13. Team Emotion	0.0100	0.0420	0.066 **	−0.0340	−0.085 ***	0.085 ***	−0.062 **	0.097 ***	0.0410	0.00300	−0.0380	0.173 ***	1

Note: *** *p* < 0.01, ** *p* < 0.05, * *p* < 0.1.

**Table 7 ijerph-20-01915-t007:** Results of the regression model—patients’ emotional well-being (n = 1318).

Hypothetical	Variables	Model 1 Basic Model	Model 2 Team-Level	Model 3 Individual-Level
Control				
	Team Price	0.009 **	0.009 **	0.01 **
		(0.048)	(0.036)	(0.015)
	Team Longevity	−0.037 **	−0.041 **	−0.031 *
		(0.027)	(0.014)	(0.057)
	Team Size	0.001	0	0.002
		(0.695)	(0.958)	(0.495)
	Reply Rate	−0.008	−0.007	−0.011
		(0.647)	(0.694)	(0.537)
	Team Help	0.003	0.005	0
		(0.442)	(0.225)	(0.962)
	Team Comprehensive Level	−0.001	−0.004	0
		(0.857)	(0.604)	(0.998)
	Leader Involvement Ratio	−0.004	−0.002	−0.015
		(0.757)	(0.856)	(0.248)
	Disease Seriousness	−0.034 **	−0.035 **	−0.011
		(0.025)	(0.02)	(0.454)
	Patient Health Terms	−0.145	−0.341 ***	−0.28 **
		(0.185)	(0.004)	(0.012)
Main Effects				
H1a	Team Richness		0.001	
			(0.839)	
H2a	Team Perception		10.498 ***	
			(0.001)	
H3a	Team Bio		0.488 ***	
			(0.001)	
H4a	Team Emotion		0.17 ***	
			(0)	
H1b	Leader Richness			0.003
				(0.297)
H2b	Leader Perception			0.585
				(0.101)
H3b	Leader Bio			0.069
				(0.503)
H4b	Leader Emotion			0.137 ***
				(0)
H1c	Non-Leader Richness			0.001
				(0.829)
H2c	Non-Leader Perception			0.671 **
				(0.031)
H3c	Non-Leader Bio			0.358 ***
				(0.001)
H4c	Non-Leader Emotion			−0.421 ***
				(0)
	Constant	0.241 **	0.117	0.295 ***
		(0.02)	(0.26)	(0.004)
	R-squared	0.011	0.044	0.084

Note: *p*-values are in parentheses, *** *p* < 0.01, ** *p* < 0.05, * *p* < 0.1.

## Data Availability

Data available on request.

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
