# Peer review of "How Do Team-Level and Individual-Level Linguistic Styles Affect Patients’ Emotional Well-Being—Evidence from Online Doctor Teams"

_ijerph, 2023, doi:10.3390/ijerph20031915_

Round 1

Reviewer 1 Report

Please review in particular my comments regarding the practicality and effectiveness of the physician-team model with its use of simultaneous text input to patients as it appears in this paper. As you have presented it, I have not myself witnessed this model in use.

I also wish to point out that patients have varying degrees of medical knowledge and medical terminology. The model as you presented it does not appear to take into account the need fo adjust physician communications with patients.

It did not appear from what you presented, patient autonomy in health care decision-making was taken into account as you wrote about physcian dominance in physician-patient communications. If you consider and effective patient consent process as a two-way communication process, it was not clear to me how the way you presented the team-doctor, online text communication process facilitates this type of communication.

Introduction

The following line analysis includes both writing mechanics and content aspects. It does not include an analysis of the data analytic methods and I have not analyzed formatting in the reference list

Line 1 Remove “the”

Line 8 Remove the word “the” in “Using the on line doctor teams…”

Line 9 Is context really the right word?

Line 10  Change “affecting” to affects. I will be looking to see if the doctor-patient communication relationship is consistently utilized as affecting patients’ emotional well-being.

Line 11 I did not really understand of how physician-leaders and non-leaders are separately defined. This is not only true for the reader, but how to distinguish the two for the patient might also be difficult. It as been my understanding or perhaps understanding, the physician is the leader. The patient many also have communications with a referred specialist physician, or a resident/fellow, a nurse practitioner, physician assistant or other ancillary or supportive staff. Perhaps the difference between a physician leader versus a physician non-leader will be more evident to the reader and I wonder at this point how the difference is conveyed to the patient.

Lines 19-20 I did not understand why the sentence beginning with “In terms of team roles” has validity. Again, perhaps this will become evident later.

Lines 20-22 The conclusion does not relate to patients’ emotional well-being, but instead now introduces “creating high-quality doctor team services.” Prior, you had a focus on emotional well-being (line 10), and exploring the roles “played by doctors….in doctor-patient communication” (line 11).

Line 26 The sentence beginning with “Online consultation” does not appear to be grammatically correct.

Line 29-30 “an important sources” is grammatically incorrect.

Line 30 I did not really understand the meaning of the sentence beginning with “Online communication”

Line 35 Regarding the sentence beginning with “Health outcome:” In my mind attaining/achieving health is more than disease control and emotional well-being. Isn’t there a major role that online consultation can play with regard to health maintenance, prevention and well-being. I would also think that in many instances physician participation may not be needed for some of these services.

Lines 45-46 The sentence did not make sense to me as written. I do agree that online doctor teams are “a new model.” It is a model which frankly I have not seen. Perhaps I am not well-informed?

Line 53 Beginning with “it is especially important” appears to begin a run-on sentence.

Line 54 I wondered in such a team mode how you “collaboratively guarantee” what you assert.

Lines 56-59 You have introduced a model of physician team to patient communication, which you state is performed at the same time, and yet is done mainly through texting. I wondered the extent to which this is really a patient-satisfying model and practically feasible as patients must have difficulty trying to understand/comprehend written input simultaneously from multiple physician sources and then with all of this input be able to meaningfully respond in a manner which would result in appropriate patient compliance. If all of the team are sending communication at the same time, what if there is a disagreement about either diagnosis or treatment plan? How is this resolved?

Line 63-65 There have been many studies involving patient satisfaction in relation to physician communication. The HCAHPS survey is just one example in which this is tested. Linking the communication to emotional well-being can be seen with just the small following sample.

·        https://www.researchgate.net/publication/313812023_Pathway_Linking_Patient-Centered_Communication_to_Emotional_Well-Being_Taking_into_Account_Patient_Satisfaction_and_Emotion_Management

·        https://www.ncbi.nlm.nih.gov/pmc/articles/PMC2794010/

·        https://www.mdpi.com/2077-0383/6/3/33

·        https://msurgerytest.rcsi.com/wp-content/uploads/2019/09/How-Does-Communication-Heal.pdf

I therefore question the assertion made on these lines.

Lines 85-95. After reading through reference #11 which is cited in support of the paragraph, I did not see much of what was provided in the article to support some of the themes contained in these lines.

Line 89 Must online physician teams come from different hospitals as is indicated on that line?

Line 94 See my prior comments regarding multiple physicians texting to the patient at the same time.

Lines 94-103 I wondered how all of this is accomplished to effectively meet patient care needs if all physician-team members are providing input to the patient at the same time.

Lines 106-the beginning of 108. As written the sentence did not make sense to me as written. So for example what does “dominated by doctors” really mean. I thought effective communication is a two-way process. Think about the informed consent process and how the patient and not the physician is the ultimate decider about a course of treatment. Think about the concept of having the patient as a partner in decision-making, often considered essential for meaningful compliance with treatment plans.

Line 106, the last word, “and” does not belong there. 

Lines 106-116 I agree with the major themes contained in this section.

Line 126 The word “the” does not belong in the sentence.

Lines 126-135 It is interesting this section writes about satisfaction derived from the doctor-patient relationship, using the singular for doctor as opposed to the model under study, which is multiple doctors communicating simultaneously.

Lines 182-185 You state the study examines on an individual and teams level, but then only describe the individual level.

Lines 195-203 I wondered whether health literacy of patients was considered as a variable , as the section speaks of “the use of richer vocabulary.” I would think the ability of patients with variation  regarding health literacy would be important for their comprehension, particularly as all are inputting information at the same time, expecting patients’ understanding. I also did not really understand why “a lower level of linguistic diversity: would “negatively affect teams’ credibility and influence.

Lines 217-218 How is “a more generic health language” achieved with a multi-specialty doctor team, again communicating via text to the patient simultaneously. Is there a mechanism to assess the patient’s health literacy, which would also require different wording to ensure patient understanding. How does the information regarding a patient’s health literacy communicated to the team and of more importance to what extent does what is communicated to the patient adjusted to ensure patient understanding?

Line 226 “The” needs to be added before the word “doctor.”

Lines 227-240 How does the patient know and is it important for the patient to know who are the leaders and non-leaders? I am still not really understanding the different communications that are expected from each of the two groups if all are sending input to the patient simultaneously. What is the mechanism for the patient to practically understand all this input? In the consent process for example, which is a relationship between one doctor and one patient, the physician has the responsibility of communicating at a level, which the patient can understand, sometimes referred to as “in layman’s terms.” How is this addressed with the team online model?

Line 242, The word “the” prior to the word “health.”

Line 266 More than a comma is needed after the parentheses. Otherwise, this is a run-in sentence.

Line 175 Remove the word “the.”

Line 386 More than a comma is needed after the word “well being.” As is , this is a run-on sentence.

Line 388 I would capitalize “Team Longevity and Disease Seriousness”

Lines 385-392. I did not really understand the bases for the reasons why the data is indicating the stated relationships.

Lines 423-431 I have mentioned before (and perhaps I missed it), how is the patient informed of who is the “leader.” And if the patient does not know, how does this affect identification of categories and data analysis

Reviewer 2 Report

Thank you for the opportunity to review this manuscript. I have a few comments for your consideration. I hope these suggestions help.

§  The definition of “online doctor teams” can be mentioned earlier in the Introduction section, instead of the Methods section.

§  It is not so typical for an article to have such a long section on literature review. The literature review can be more concise and incorporated into the Introduction  section.

§  If there is a dedicated section on detailed findings from a literature review, it may be useful to describe the search strategy, including databases, grey literature, and key words and MeSH terms used.

§  The sections 2.1 to 2.4 and 3.1 to 3.3 are interesting. However, they are very long and it may be hard to understand what the intentions are for these sections?

§  The Methods section does not begin until Section 4. It feels like this article can be broken into 2 separate articles.

§  Unfortunately, table 3 is hard to understand and seems a bit abrupt to include it in the paper.

§  In section 2.4, the 3 research gaps seem broad and quite distinct from each other. It may help to rephrase each point as research question. If the research questions remain broad and quite different from each other, it may be beneficial to reduce the scope of this paper.

§  It may be a little challenging to follow Sections 3, 4 and 5. A suggestion could be to structure the hypothesis, methods and findings according to the 3 research questions you have. It will then be easier to understand your hypothesis, methods and results for each of the 3 different research questions.

§  The Methods section can include subsections on the sample population, intervention, etc.

§  For section 4.2.3: How were the covariates selected?

§  Some readers may argue that there will still be quite a bit of confounding. How did you address that?

§  For section 4.2.4: Do you mean “multivariable”?

§  The section on data analysis may be a bit vague. Why was linear regression selected? Were model assumptions tested?

§  There are at least 3 models describes. While this is comprehensive, it may be challenging for practitioners and policymakers to decide which results are the most applicable to their situations. Further explanation of the implications of the different modes will help.

§  The results did not have sample characteristics of the patients and online doctors.

§  Similar to what was shared above, it is hard to follow how the results address the 3 research questions listed in the earlier sections. The findings are be structured according to the 3 research questions.

§  Table 5: Column on sample size can be removed. Include n=x in title.

§  Further elaboration is needed on how to interpret the 3 tables in the Results section.

§  Unfortunately, there is no Discussion section, which it quite crucial.

§  What are the implications of the findings, comparison with literature, strengths and limitations?

§  After reviewing the article, the title may not be the most appropriate. Maybe the exposure and outcome can both be mentioned in the title. The methodology and/or sample population could also be mentioned.

§  The conclusions in the main text and abstract may be a little too exaggerated and do not reflect the findings. Again, the conclusions need to show how the 3 research questions were addressed. The article may benefit from reducing the scope of the research and break this up into 2 papers.

§  In section 4.2.1, it says “variables”, but there is only 1 paragraph on a single variable in italics.

§  Some proofreading is needed for minor edits.

Round 2

Reviewer 1 Report

A study Limitation: The authors should indicate in their paper, something with regard to the on-line medical team model, as one that has emerged in China, but does not appear to have been adopted in Western countries. It is unclear as to the extent of adoption in China.

            Comments regarding specific elements from the revised manuscript:

Line 12.

It is unclear how the terms doctors who are leaders versus non-leaders are distinguished.  Some clarification is required. In American terms for example, how does this relate to an “attending” versus a “resident” or “intern.” Is there more than one attending in a team and if so, which is the “lead?”

Lines 20-21. 

“Leaders should focus on the emotional expression, whereas non-leaders should focus on the use of perceptual and biological words.” Again, the dichotomy of leaders versus non-leaders is seen here.

Again, What are the criteria for each?

Lines 47-51.

“An online doctor team is usually led by a well-known expert in the field [9]. 47 General doctors and young doctors join in and establish a collaborative team to provide consultations and treatment services for patients [10]. In a team, doctors could assign consultants by considering their availability to provide quick health service and discuss the medical cases with fellow doctors to guarantee the accuracy of diagnosis [11].”

There is remains uncertainty as to the classification of physicians on a team. There is the “well-known expert” (In the U.S. models a recommendation would likely be made by a primary care physician or in managed care models, a referral to a specialist, in-network would be typically required. In non-managed care models, a self-referral to a specialist can be acceptable without prior approval from a PCP.). The term “general doctor” now appears, and along with “young doctors,” what both represent is unknown. There are now also  here “consultants” and “fellow doctors.” Which represent the leaders and the non-leaders. Which communicate directly with patients?  

Lines 76-77

[Research] “Question two: Are there different roles for leaders and non-leaders in the online doctor teams in the improvement of patients’ emotional well- being through online interactive communication?”

Clarification of who is a leader versus who is a non-leader is required, as indicated with in prior comments.

Lines 107-108

“Patients can search for appropriate medical doctors, describe their symptoms to them, and upload the related medical test results.”

If a team is led by “a well known expert in the field,” how is a patient to determine who is “a well known expert?” How is a patient to determine who else should be on the medical team? How is the medical team put together, i.e. by the patient. Of interest too, is the role of the patient to upload “related medical test results.” In actuality what is the role of the patient in the physician selection process?

Lines 171-172

“We also investigate the different roles of team leaders and non-leaders when participating in team consultations.”

Clarification of terms as previously mentioned is required.

Lines 186-194

“On online consultation platforms, the registered doctors may differ in the level of vocabulary richness they use to provide services because of their diverse professional backgrounds, education levels, and work experiences. In health care, doctors who are familiar with the names of various diseases, symptoms, drugs, treatment methods and body parts may express their recommendations effectively, thus enhance the efficiency of communication and the outcome of consultation services. It has been shown that in online teams, the use of richer vocabulary during communication can engage audience [44]. Therefore, it is expected that vocabulary richness has a positive effect on patients’ emotional well-being when online doctor teams consult with patients.”

The utility of the term “vocabulary richness” is not really clear here.   A licensed physician, irrespective of background and work experience, if placed on a team would have a rich medical background, certainly far exceeding that of the majority of their patients. In fact, the disparate nature of knowledge and vocabulary between a physician and a patient can serve as a serious communication barrier. In the consent process in the U.S., we speak of the need to communicate with the patient, using language the patient can understand. This is congruent with what appears in the following section, which begins on line 204.

Reviewer 2 Report

I have no further comments. Many thanks.

Author Response

Thanks for your help for revising the paper. Many thanks.